# Acceptability and usability of a mobile application for management and surveillance of vector-borne diseases in Colombia: An implementation study

Sarita Rodríguez[1☯]*, Ana María Sanz[1☯], Gonzalo Llano[2‡], Andrés Navarro[2‡], Luis Gabriel Parra-Lara[1,3☯], Amy R. Krystosik[4‡], Fernando Rosso[1,3,5☯]

1 Centro de Investigaciones Clínicas (CIC), Fundación Valle del Lili, Cali, Colombia, 2 Grupo de Investigación i2T, Universidad Icesi, Cali, Colombia, 3 Facultad de Ciencias de la Salud, Universidad Icesi, Cali, Colombia, 4 Division of Infectious Diseases, Department of Pediatrics, School of Medicine, Stanford University, Stanford, CA, United States of America, 5 Infectious Diseases Service, Department of Internal Medicine, Fundación Valle del Lili, Cali, Colombia

☯ These authors contributed equally to this work.
‡ These authors also contributed equally to this work.
* sarita.rodriguez@fvl.org.co

**Data Availability Statement:** All relevant data are within the manuscript and its Supporting Information files.

## Abstract

### Background

Vector-borne diseases are a public health problem in Colombia, where dengue virus infection is hyperendemic. The introduction of other arboviruses, such as chikungunya and Zika in the last three years, has aggravated the situation. Mobile health (mHealth) offers new strategies for strengthening health care and surveillance systems promoting the collection, delivery, and access of health information to professionals, researchers, and patients. Assessing mobile application performance has been a challenge in low- and middle-income countries due to the difficulty of implementing these technologies in different clinical settings. In this study, we evaluate the usability and acceptability of a mobile application, FeverDX, as a support tool in the management of patients with febrile syndrome and suspected arboviruses infection by general practitioners from Colombia.

### Methods

A pilot implementation study was conducted to evaluate the usability and acceptability of FeverDX using the modified version of the Mobile Application Rating Scale (uMARS). The evaluation form included 25 questions regarding quantity and quality of information, engagement, functionality, aesthetics, impact, and acceptability by healthcare workers. Each item uses a 5-point scale (1-Inadequate, 2-Poor, 3-Acceptable, 4-Good, 5-Excellent). A global score was obtained for the evaluation form test by determining the median scores of each subsection. A descriptive statistical analysis of the data obtained was performed.

### Results

Between December 2016 and January 2017, a total of 20 general practitioners from the Emergency room and hospitalization areas evaluated FeverDX. Less than half (9/20) of the

**Funding:** This study was funded by Fundación Valle del Lili and Universidad Icesi. The funders don't play role in the study design, data collection and analysis, decision to publish, or preparation of the manuscript.

**Competing interests:** The authors have declared that no competing interests exist.

evaluators had a comprehensive knowledge of the Colombian Ministry of Health's guidelines for the diagnosis and management of arboviruses, and evaluators partially (4/9) or completely (5/9) agreed that the content of the application follows the management guidelines. On uMARS scale, FeverDX excelled regarding impact (median 5; IQR = 5–5), functionality (median 5; IQR = 4.8–5), and information and scientific basis (median 4; IQR = 4–4). FeverDX scored well regarding user feedback (median 4; IQR = 4–4.5), design and aesthetics (median 4; IQR = 4–4.3), and subjective assessment of quality (median 4.5; IQR = 4.3–4.8).

## Conclusions

FeverDX, a mobile application, is a novel mHealth strategy to strengthen care processes and facilitate the detection and reporting of notifiable surveillance diseases. It could improve adherence to clinical practice guidelines for the management and prevention of prevalent diseases as arboviruses in healthcare settings.

Although this pilot study used a small sample size, FeverDx performed adequately in a simulated emergency consultation. Further implementation studies are needed to increase the reliability of mHealth technologies in different scenarios.

## Introduction

In Latin America and the Caribbean, arboviruses are a major public health problem [1]. Before the arrival and subsequent spread of chikungunya and Zika viruses, *Aedes* spp. transmitted dengue virus with a record of 2,38 million cases in the Americas in 2016 [2]. During 2014, Colombia experienced the importation and rapid transmission of chikungunya, with approximately 96,687 cases reported. In September 2015, Colombia reported its first case of Zika, reaching a total of 11,712 the same year [3].

The limited availability of diagnostic tests in endemic areas in developing countries, represents a challenge for differential diagnosis, especially in the case of co-infection [4–6] because arboviruses infections share non-specific clinical manifestations [7]. This situation encourages the development and implementation of innovative initiatives in different scenarios, as health services and public health surveillance entities. An example is mobile health (mHealth), referring to the practice of medical and public health supported by mobile devices, such as mobile phones, patient monitoring devices, personal digital assistants, and other wireless devices [8]. For more than a decade, the World Health Organization (WHO) has promoted the development and application of mHealth as an essential resource in the provision of health services [9]. According to a survey conducted by WHO, only 12% (14/115) of the WHO Member States conducted formal evaluations of mHealth-related initiatives [8].

Supply and demand for mobile applications development in the healthcare area have been recently increasing [10]. Although mHealth performance evaluations are growing [11–13], regarding the majority of commercially mobile health applications in our research field, companies and researchers have not yet incorporated an extensive performance evaluation, including usability and app quality. Furthermore, additional tailored and rigorous evaluation studies on the mHealth clinical impact and reliability are still required [14,15].

FeverDx is a mHealth tool designed by a group of physicians for healthcare workers (HCW) in a local context where digital transformation is limited, making this tool tailored to

the specific needs of the end-user in terms of time and ease of use, access to national health guidelines content, and is a critical step of the reporting process of mandatory notifiable diseases. Moreover, the mobile app provides rapid access without an internet connection to locally stored clinical data regarding the most frequent fever syndromes clinical guidelines and supports general practitioners (GPs) in the approach and management of patients with a fever syndrome and suspected arboviral infection.

This study evaluated the usability and acceptability of FeverDx mobile application, a mobile application to guide general practitioners (GPs) in the approach and management of patients with a fever syndrome and suspected arbovirus infections. GPs from different clinical areas at Fundación Valle del Lili (FVL) performed FeverDx.

## Materials and methods

We conducted a cross-sectional, pilot implementation study of FeverDx in Cali, Colombia.

### Setting

The study was conducted at Fundación Valle del Lili (FVL) University Hospital between December 2016 and January 2017. FVL is a non-profit university hospital that works as a referral health center for the southwestern region of Colombia, and it is associated with Universidad Icesi. In 2016, a total of 17,321 cases of dengue, 1,204 of chikungunya, and 14,638 of Zika were reported in the city by the Local Health Authority [16].

The study protocol was approved by the Comité de Ética en Investigación Biomédica from FVL (IRB/EC No. 584–201, followed the ethical principles for medical research outlined by the Declaration of Helsinki [17] and the declaration 8430 of 1993 from Colombia's Ministry of Health [18].

### Mobile application

FeverDx is a mobile application (app) developed by the FVL in association with Universidad Icesi. The app was created following the Methodology for the Development for Mobile Applications (MDAM) [19]. This guideline recommends five phases intending to guide the process of app design and implementation: analysis, design, development, performance testing, and delivery. The minimal requirements for the use of the app are a tablet or smartphone device supported by either Android and iOS operating systems with at least intermittent access to Wi-Fi or mobile network connectivity.

This app runs based on two modules: the clinical guideline and the evaluation modules. The guideline module allows GPs to obtain a rapid-offline reference to the CPGs for acute febrile syndromes such as Zika, dengue, chikungunya, and other common infections in tropical areas (leptospirosis). Each guideline includes general topics concerning clinical features, diagnostic, treatment, and prevention (Fig 1).

The patient evaluation module allows the GP to create a local record (mobile phone database) of the patient data, which can be consulted, modified, or deleted at any time. Fig 2 shows the algorithm integrated into the application to support the approach and management of patients with a febrile syndrome. This algorithm highlights the essential points of patient care, such as the evaluation of general symptoms, vital signs, and the alarm signs, severity signs, which include all dengue, Zika, and chikungunya warning signs.

Patient information obtained from medical history, physical examination, and laboratory tests are recorded using a graphical interface (checklists and alphanumeric fields) (Fig 3).

The mobile app shows a variety of messages from an internal feedback "library" which is based on the Colombian Ministry of Health´s clinical practice guidelines (CPG) for the

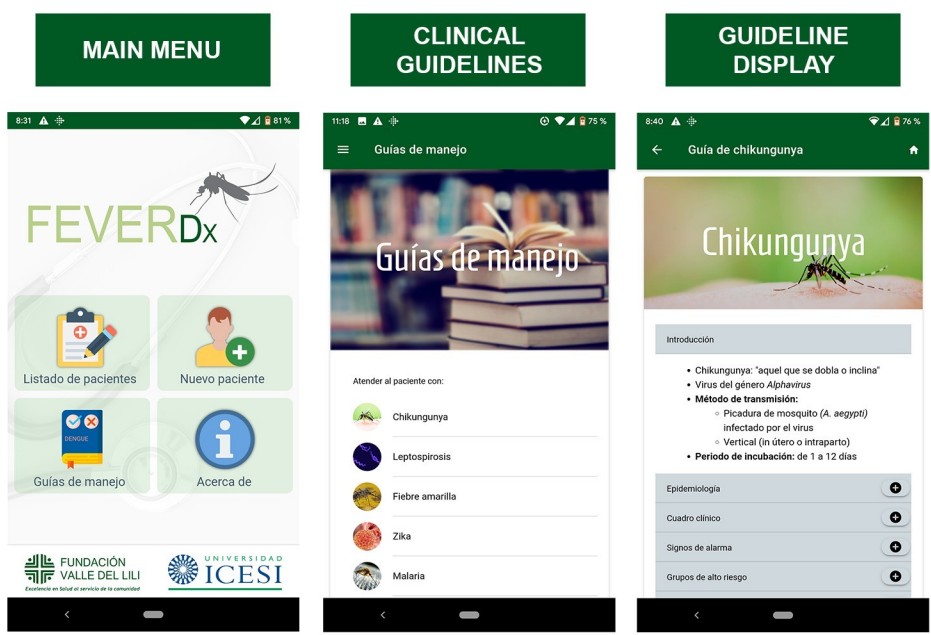

**Fig 1. Screenshots of the FeverDx clinical guideline module.** A. Main Menu; B. Clinical Guidelines; C. Guideline Display.

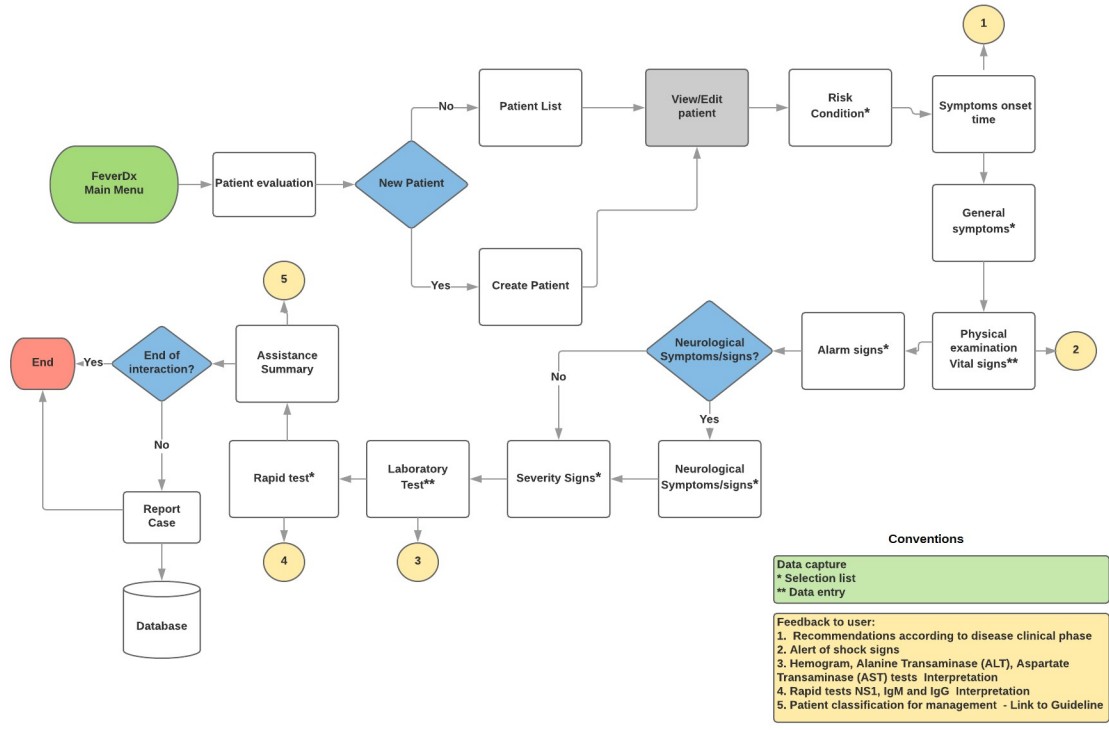

**Fig 2. Flowchart of the patient evaluation module process.**

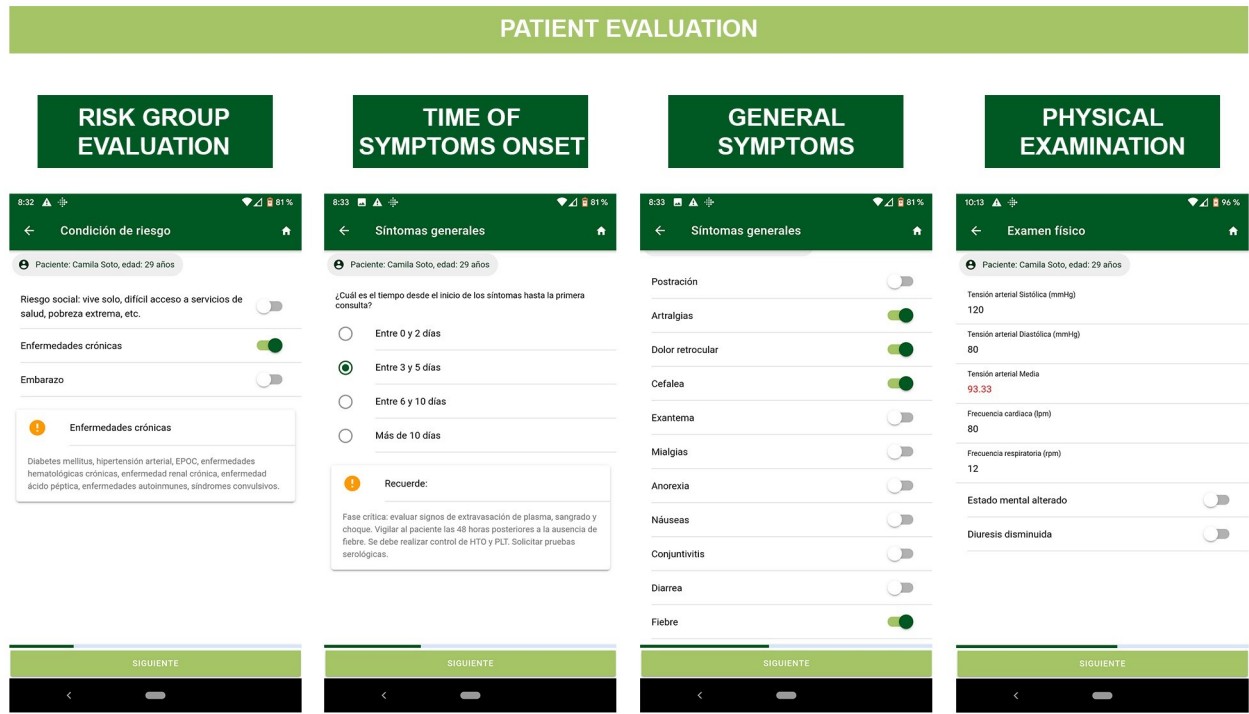

**Fig 3. Patient clinical information collection views in FeverDx application.** A. Risk Conditions; B. General Evaluation; C. Clinical Evaluation; D. Physical Exam.

diagnosis and management of arboviruses (dengue, chikungunya, and Zika) and additional common infections in tropical countries [20–23]. An infectious disease specialist reviewed the included recommendations.

Obtained data from registration, evaluation, and reporting is stored on a local server, where it is available to be consulted by national epidemiological surveillance entities.

The FeverDx app is compliant with the Policy of treatment and protection of personal data (apps.co) from the Colombian Ministry of Information and Communications Technologies [24].

## Study procedures

The pilot study was conducted in a controlled setting where GPs from the emergency room and hospitalization areas, that had no prior experience with the app, were recruited through an individual invitation. The usability evaluation was carried out in a single session according to the following stages:

1. Delivery of materials (an Android or iOS device, medical records, evaluation forms) and instructions about the application use and the evaluation process.

2. Simulated emergency consultation (Interaction with the application).

3. Individually and confidentially usability evaluation by GPs.

The convenience sampling method was implemented to select a sample of 100 different clinical records of patients who consulted FVL for acute febrile illness from the Institutional Epidemiological Surveillance Committee and Statistics Department Databases. Demographic, clinical, and laboratory data were obtained from the FVL medical charts. Five clinical records

were randomly assigned to each of the twenty GPs who used the app to perform a simulated emergency consultation interacting with the patient evaluation module.

During the clinician-application interaction, feedback messages related to patient risk-group, signs of alarm, severity, and recommended management, appear intended to guide the GP in the approach and management of the patient.

The simulated emergency consultation that included the entry of all patient-related data, lasted 6 minutes on average. Afterward, there was no time limit to experience FeverDx freely. The evaluation of usability and acceptability was performed anonymously, individually, and secretly by each GP. The assessment included the perception of usefulness and user experience satisfaction.

## Evaluation tool and analysis

Implementation outcomes were extracted into a template using a Microsoft Excel® spread-sheet. We developed the evaluation form based on four different evaluation systems for mHealth applications: 1) mobile health (mHealth) evidence reporting and assessment (mERA) checklist [25]; 2) iSYScore index [26]; 3) Mobile App Rating Scale (MARS) [27]; 4) User Version of the Mobile Application Rating Scale (uMARS) [28]; and 5) additional evaluation methods [29] and implementation outcomes (acceptability [30] and usability [31]).

The evaluation form comprises 25 questions measuring quantity and quality of information (Section A), engagement (Section B), functionality (Section C), aesthetics (Section D), the impact of patient management from the practitioner perspective (Section E) and acceptability by HCW (Section F) (S1 File).

Knowledge evaluation regarding the CPG for vector-borne disease management was included in the first two questions of the evaluation form. Each item uses a 5-point scale (1-Inadequate, 2-Poor, 3-Acceptable, 4-Good, 5-Excellent) (S1 File). A global score was obtained for the evaluation form test by determining the median scores of each subsection.

A descriptive analysis was performed using STATA® statistical software (Stata Corp, 2011, Stata 12 Base Reference Manual, College Station, TX, USA). Continuous data were summarized as median and interquartile ranges, and categorical variables presented as frequencies and proportions.

## Results

Between December 2016 and January 2017, a total of twenty GPs tested and evaluated FeverDx mobile application.

## CPG knowledge

All data regarding CPG knowledge evaluation was gathered from GPs with no prior use of the FeverDx app. The proportion of GPs with comprehensive knowledge of the Colombian's CPG for diagnosis and management of arboviruses (dengue, chikungunya, and Zika) was less than half of GPs evaluated (9/20), and less than one-third of them applied them in their practice (6/20). Of those GPs who reported having comprehensive knowledge of CPG, more than half (5/9) wholly agreed and, 4/9 partially agreed that the application information matched with CPG information.

## Application evaluation

Table 1 presents an overview of the information quality, engagement, functionality, aesthetics, quality, and impact categories evaluation scores. The median of the global evaluation score

**Table 1. Category and item scores evaluated in the FeverDx application (n = 20).**

| Category | Score Median (IQR) |
|---|---|
| **Information/scientific and clinical foundations** | **4 (4–4)** |
| Implementation has specific, measurable and achievable objectives | 4 (3–4) |
| Quality of information | 4 (4–5) |
| Quantity of information | 4 (4–5) |
| Visual information | 4 (4–4.3) |
| Credibility | 4 (3–4) |
| References | 4 (3.5–5) |
| The application can increase your knowledge/ comprehension about vector-borne diseases approach and management | 5 (4.7–5) |
| I had to learn many things before I could use the application | 2 (1–2) |
| **User feedback** | **4 (4–4.5)** |
| Interest | 5 (4–5) |
| Interactivity | 4 (4–5) |
| Application content is appropriate for the end user | 4 (4–5) |
| **Functionality** | **5 (4.8–5)** |
| Throwput | 4 (4–5) |
| Ease of use | 5 (4–5) |
| Navigation | 5 (4–5) |
| Gestural Design (Interactions: Taps/Swipes/Pinches/Scroll) | 5 (4–5) |
| **Aesthetic—graphic design** | **4 (4–4.3)** |
| Design | 4 (4–5) |
| Graphs | 4 (4–5) |
| Visual appeal | 4 (4–5) |
| **Impact** | **5 (5–5)** |
| This application can increase motivation/intention to improve adherence to handling guidelines of dengue, Zika, and Chikungunya | 5 (4–5) |
| Attitudes towards the approach and management of patients with dengue, Zika, and chikungunya are likely to change. | 5 (4–5) |
| This application has the potential to be efficiently introduced into multiple emergency departments of different levels of care in Colombia | 5 (4.8–5) |
| **Acceptability** | **4.5 (4.3–4.8)** |
| I would recommend this application | 5 (4–5) |
| Would frequently use this application | 4 (4–5) |

IQR: Interquartile Range

The highest score is five and the lowest score is one, except in section 1, item 8, where the highest score equaled one, and the lowest score equaled 5.

was 4.0 (out of 5.0). Impact and functionality obtained the highest score with a median of 5 (IQR = 5–5) and 5 (IQR = 4.8–5), respectively. Among GPs, 19 out of 20 partially/wholly agreed that the application could increase their knowledge and understanding about dengue, chikungunya, and Zika infection management.

Regarding the user feedback, GPs scored a median of 4 (IQR = 4–4.5), and "interest" obtained the high scored of this category 5 (IQR = 4–5). Almost one-third of participants (6/20) reported the mobile application content as suitable for physicians (without content problems) and more than half (13/20), that the application information (content) was correctly oriented with minor issues. Only one of the participants considered the content as acceptable and it was not geared toward physicians.

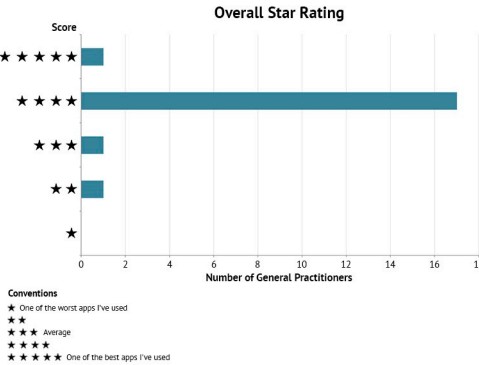

**Fig 4. Overall star rating of mobile application FeverDx.**

Concerning impact, the intention to change domain was assessed with a median score of 5 (IQR = 4–5), and the majority (18/20) of GPs partially/wholly agreed that the app will probably change attitudes towards clinical management of dengue, chikungunya, and Zika. Moreover, all the GPs would recommend the app, and most of them (17/20) rated it with 4 out of 5 according to the overall star rating (Fig 4).

## Discussion

To our knowledge, this is the first study evaluating the usability and acceptability of a mobile application for the clinical approach to acute febrile syndrome management in Colombia. In general, there is a lack of evidence regarding the assessment and performance of mobile apps in healthcare facilities. For this reason, more initiatives to evaluate mHealth developments are needed in different contexts to know the real impact of these technologies, and to generate evidence about its performance in the field [15].

In this study, we described an alternative approach to evaluate mHealth through validated scales such as uMARS, which had demonstrated a reliable degree of app quality evaluation and the ability to obtain valuable information from users about mobile apps [28]. Our findings showed that FeverDx evaluation obtained a high overall score as well as in most of its subscales. Hence, the use of FeverDx in regions where arboviruses are endemic or hyperendemic potentially could help GPs and stakeholders in the management of acute febrile syndromes such as arboviruses infections. Although this is a pilot study, our findings are a starting point for the development of new implementation studies that allow the evaluation of application performance in different contexts and scenarios.

One of the challenges in low and middle-income countries is the surveillance of vector-borne diseases and their epidemiology. The epidemiological surveillance process includes the notification, collection, and analysis of data, which allows for the generation of timely, valid and reliable information for stakeholders and policymakers, and thus to guide public health policy and create prevention and control strategies [32]. However, this process is often time-consuming. We believe that FeverDx could help in the solution of this problem and facilitate this process through innovative development that is easy to reach and manage from an app that can be used in devices such as smartphones.

Mobile applications designed for epidemiological surveillance of arboviruses are limited in their scope. A study conducted in 2018, which reviewed the available Google Play and App Store applications, showed that 26 applications relevant to surveillance are available. This review included 7 surveillance applications related explicitly to dengue and Zika, aimed at case

reporting and outbreaks monitoring through the use of interactive maps. Some of these applications were limited by location, operating system (Android only), and although most of them were developed for global use, most of them were in the English language, which could narrow their applicability in different contexts [33,34]. Additional mHealth developments with a different approach are available such as surveillance (*DengueWatch*, *Dengue map*, *ZikaTracke*, *Kidenga App*, *Break Zika*, *Predict and Beat dengue*) and general information (*Dengue–Manejo clínico*, *Arbo App*, *Dengue App*, *Dengue síntomas tratamiento y prevención*, *Dengue Treatment*, *WHO Zika App*), however evidence regarding implementation in clinical scenarios is scarce [34,35].

There are relevant conditions to be considered for the mobile health applications development and adoption: the local demographic, geographic, connectivity, culture, and socio-sanitary aspects; and the particular needs of the potential end-users, general public, healthcare workers or surveillance authorities [34,36]. Among the commercially available mHealth developments, a small proportion combines different functionalities such as vector-borne disease management (evaluation, classification, and treatment) or rapid offline clinical information review.

In this sense, FeverDx is innovative in the field of arboviruses, it is in the local language, and has been evaluated for usability and acceptability. It offers a solution to local and national surveillance systems, based on these identified needs, to fulfill an early and timely case report. Furthermore, it could potentially support the decision-making process in real-world clinical environments.

Another relevant finding was that a not insignificant proportion (25%) of the GPs had not read the Colombian CPGs for the management of vector-borne diseases. Previous studies in this matter have shown limitations in the adoption of CPGs. For example, a study conducted in Zimbabwe in 2011 that evaluated the attitudes of GPs towards the use of CPGs showed that 57.9% of GPs felt that guidelines would not improve their diagnostic ability, even though that 65.9% were prepared to use guidelines in their practice [37]. Furthermore, qualitative studies conducted to assess GPs' experiences when applying multiple guidelines found that GPs' responses clustered around two major topics: 1) Complications for the GPs of applying multiple guidelines; and, 2) Complications for their patients when GPs apply multiple guidelines [38]. Thus, further strategies for CPGs implementation and real-world use are needed. For this reason, we propose friendly mobile technologies as a feasible and reliable strategy for promoting the adoption of CPGs.

Finally, mobile technology offers unique opportunities to reach and engage HCW to enhance health literacy and public health authorities for real-time disease surveillance, and that the real-time case reporting feature could improve the current manual registration and report process [33].

FeverDx is a user-centered app intended to drive clinical information to GPs and facilitate epidemiological surveillance. Our findings could promote the adoption and engagement of this app in a clinical setting; thus, improving the care of patients with febrile syndromes in health centers in tropical areas resulting, eventually, in better clinical outcomes.

## Limitations

Our study is not without limitations and results should be interpreted in the context of the study design. First, the study included twenty GPs from the emergency department and hospitalization areas of one center highly specialized in the treatment of severely ill patients, which limits the generalizability of our study and the applicability of our results in other environments. Furthermore, the GPs included in the study and the mobile application developers

come from the same institution, which may have introduced a courtesy bias. To face this limitation, we assured an anonymous and independent evaluation process for the participants. Second, the fact that we tested the app in a tertiary care center makes our study prone to selection bias. Third, the implementation of FeverDx is currently restricted to Colombia because its design was mainly based on the Colombian Ministry of Health´s mandatory notification sheets and CPG. Application content was strengthened with the international recommendations given by the WHO and with the review by an infectious diseases specialist with experience in arboviral research to mitigate this issue.

Despite our limitations, we consider that the application, in general, has acceptable performance, and our study advances the knowledge of mHealth technology for the diagnosis of febrile syndromes (arboviruses and other common infections) in settings where innovative initiatives are needed to improve patient outcomes. Finally, further research with a larger sample size and with the participation of other healthcare centers need to be conducted to support our initial findings.

## Supporting information

**S1 File. Evaluation form (English version).**
(PDF)

**S2 File. Evaluation form (Spanish version).**
(PDF)

**S1 Dataset. Raw evaluation results.**
(PDF)

## Acknowledgments

We thank the team of developers at Universidad Icesi and the editorial group at Fundación Valle del Lili for their support in this project.

## Author Contributions

**Conceptualization:** Sarita Rodríguez, Ana María Sanz, Gonzalo Llano, Andrés Navarro, Fernando Rosso.

**Formal analysis:** Sarita Rodríguez, Ana María Sanz, Luis Gabriel Parra-Lara, Amy R. Krystosik.

**Funding acquisition:** Fernando Rosso.

**Investigation:** Sarita Rodríguez, Ana María Sanz, Gonzalo Llano, Andrés Navarro, Luis Gabriel Parra-Lara, Amy R. Krystosik.

**Methodology:** Sarita Rodríguez, Ana María Sanz, Luis Gabriel Parra-Lara, Amy R. Krystosik, Fernando Rosso.

**Project administration:** Sarita Rodríguez, Fernando Rosso.

**Resources:** Fernando Rosso.

**Software:** Gonzalo Llano, Andrés Navarro.

**Supervision:** Fernando Rosso.

**Validation:** Luis Gabriel Parra-Lara.

**Visualization:** Luis Gabriel Parra-Lara.

**Writing – original draft:** Sarita Rodríguez, Ana María Sanz, Gonzalo Llano, Andrés Navarro, Luis Gabriel Parra-Lara, Amy R. Krystosik, Fernando Rosso.

**Writing – review & editing:** Sarita Rodríguez, Ana María Sanz, Gonzalo Llano, Andrés Navarro, Luis Gabriel Parra-Lara, Amy R. Krystosik, Fernando Rosso.

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
