## [Decision Letter · Decision Letter 0]

3 Jan 2020

PONE-D-19-31941

Acceptability and usability of a mobile application for management and surveillance of vector-borne diseases in Colombia: An implementation study

PLOS ONE

Dear Dr Rodriguez,

Thank you very much for submitting your manuscript "Acceptability and usability of a mobile application for management and surveillance of vector-borne diseases in Colombia: An implementation study" (#PONE-D-19-31941) for review by PLOS ONE. As with all papers submitted to the journal, your manuscript was fully evaluated by academic editor (myself) and by independent peer reviewers. The reviewers appreciated the attention to an important health topic, but they raised substantial concerns about the paper that must be addressed before this manuscript can be accurately assessed for meeting the PLOS ONE criteria. Therefore, if you feel these issues can be adequately addressed, we invite you to submit a revised version of the manuscript that addresses the points raised during the review process. We can’t, of course, promise publication at that time.

We would appreciate receiving your revised manuscript by Feb 17 2020 11:59PM. To enhance the reproducibility of your results, we recommend that if applicable you deposit your laboratory protocols in protocols.io, where a protocol can be assigned its own identifier (DOI) such that it can be cited independently in the future. For instructions see: http://journals.plos.org/plosone/s/submission-guidelines#loc-laboratory-protocols

We look forward to receiving your revised manuscript.

Kind regards,

Abdallah M. Samy, PhD

Academic Editor

PLOS ONE

Journal Requirements:

**Reviewers' comments:**

Reviewer's Responses to Questions

**Comments to the Author**

1. Is the manuscript technically sound, and do the data support the conclusions?

Reviewer #1: Partly

2. Has the statistical analysis been performed appropriately and rigorously? 

Reviewer #1: No

3. Have the authors made all data underlying the findings in their manuscript fully available?

Reviewer #1: No

4. Is the manuscript presented in an intelligible fashion and written in standard English?

Reviewer #1: Yes

5. Review Comments to the Author

Reviewer #1: The authors present the results of a pilot acceptability and usability mobile application to help GPs with diagnosing arbovirus and leptospirosis (although it is ignored in most of the text which is centered on arboviruses), and allow them to report cases directly for surveillance purposes. The paper is well written, and I found the mhealth application interesting, useful and well-though. However, the study presented here is very limited, biased and, although it is a good first step, the study needs to be expanded in order to warrant a manuscript rather than a short communication. I would suggest expanding the study to other settings and the sample size.

In the introduction, the authors need to introduce why FeverDX is particularly suitable to improve the adherence to CPGs and surveillance, who is it designed for and how are mhealth tools particularly suitable (something novel is not necessarily better). Some of this information is already in the discussion, but I think it would help construct the argument better if placed in the introduction (for example, lines 208-215 and lines 216-219). Moreover, the whole argument is centered around that performance evaluations have not been performed for mhealth interventions (lines 69-70), which is not entirely true. There is a whole body of literature that does exactly that, and although not all mhealth tools are validated, there are performance evaluations. The reference provided for that claim, does not support it exactly.

In Methods, I would suggest making the first paragraph describing the mobile app clearer. As I continued reading, I understood the app but it is very confusing at the beginning. I would make Fig 2, Fig 1 instead. However, I am most worried about the usability evaluation. Did it happen during the training session, after the simulated emergency consultation? Or some time after? Did the authors measure actual engagement after the training session? I worry that the high scores are influenced by courtesy bias, apart from the selection bias already mentioned by the authors. Besides, the sample size is too small to even speculate about the external validity of the results.

In the results:

How was CPG knowledge measure?

In lines 164-165 the authors mention that the majority of the GPs agreed that the app matched the CGP information, but in the previous sentence that almost half actually read them… so that could be also indicating courtesy bias?

In the discussion, the first sentence speaks about the novelty of the study but uses a very restrictive situation so, yes it is novel for that…

Also, I do agree that more usability and feasibility studies are needed for mhealth but is this study enough to state this? Is a one-time (not longitudinal) and with a very small sample. This limits the interpretation of the results and I worry about speculations as the one presented in lines 205-207.

In lines 219, the authors may have omitted other apps. Of the top of my head I can think of Kidenga, for example. What are the other 26 apps relevant to surveillance referring to?

Lines 236-237 seem to talk about the general public but this app is not intended for that.

Minor:

Lines 145-147: it is not very clear how the evaluation form was given in this sentence and Table 1 is presented in Results. I would suggest including the evaluation form as supp. material and improve this sentence.

Lines 150-151: I assume that the authors referred to median and IQR as is stated in the following paragraph.

Line 183: “as (un)acceptable”?

6. PLOS authors have the option to publish the peer review history of their article (what does this mean?). If published, this will include your full peer review and any attached files.

Reviewer #1: No

---

## [Author Response · Author response to Decision Letter 0]

18 Feb 2020

Reviewers' comments:

Reviewer's Responses to Questions

Comments to the Author

1. Is the manuscript technically sound, and do the data support the conclusions?

Reviewer #1: Partly

2. Has the statistical analysis been performed appropriately and rigorously? 

Reviewer #1: No

3. Have the authors made all data underlying the findings in their manuscript fully available?

Reviewer #1: No

We appreciate this observation; thus, to compliant the PLOS Data Policy, we will make raw data concerning the results of the FeverDx usability and acceptability evaluation that supports our work fully available as part of the supporting information.

4. Is the manuscript presented in an intelligible fashion and written in standard English?

Reviewer #1: Yes

5. Review Comments to the Author

Reviewer #1: The authors present the results of a pilot acceptability and usability mobile application to help GPs with diagnosing arbovirus and leptospirosis (although it is ignored in most of the text which is centered on arboviruses), and allow them to report cases directly for surveillance purposes. The paper is well written, and I found the mhealth application interesting, useful and well-though. However, the study presented here is very limited, biased and, although it is a good first step, the study needs to be expanded in order to warrant a manuscript rather than a short communication. I would suggest expanding the study to other settings and the sample size.

We thank you for this comment. We agree that in some sentences in the abstract and the introduction, it seemed that we focus on arboviruses and leptospirosis. However, we included leptospirosis and other infections in our environment due to the frequency of presentation and that these infections are part of the differential diagnosis of dengue, Zika, and chikungunya. Consequently, we replaced "Leptospirosis" from Lines 52, 185, 457 for "other common infections" in the file “Manuscript with track changes”.

We share your concerns about the extend, bias, and other limitations of our pilot study. We agreed that our study could be perceived as a first approach, especially in a sanitary context, and that lacks evidence-based mHealth interventions. We believe that these findings warrant further research as the second phase of this study that includes a larger sample size and different participating centers.

In the introduction, the authors need to introduce why FeverDX is particularly suitable to improve the adherence to CPGs and surveillance, who is it designed for and how are mhealth tools particularly suitable (something novel is not necessarily better). Some of this information is already in the discussion, but I think it would help construct the argument better if placed in the introduction (for example, lines 208-215 and lines 216-219). 

We appreciate this suggestion. We have edited the introduction section and included the following text in lines 123-131 of the file “Manuscript with track changes”:

“FeverDx is a mHealth tool designed by a group of physicians for healthcare professionals in a local context where digital transformation is limited, making this tool tailored to the specific needs of the end-user in terms of time and ease of use, access to content and particularly in a critical step of the reporting process of mandatory notifiable diseases. This particular feature could potentially improve the currently manual registration and report process, mostly using physical resources (paper forms, physical files)[10]. Moreover, the mobile app provides rapid access without an internet connection to locally stored clinical information regarding the main topics of frequent fever syndromes clinical guidelines”.

Moreover, the whole argument is centered around that performance evaluations have not been performed for mhealth interventions (lines 69-70), which is not entirely true. There is a whole body of literature that does exactly that, and although not all mhealth tools are validated, there are performance evaluations. The reference provided for that claim, does not support it exactly.

We thank the reviewer for pointing out that we needed reference support about performance evaluations for mHealth interventions. We have added additional sources (Lines 133 and 153) and edited Lines 111-131 in the “Manuscript with track changes” file with the following text. 

Although mHealth performance evaluations are growing[12–14], regarding the majority of commercially mobile health applications in our research field, companies and researchers have not yet incorporated an extensive performance evaluation, including usability and app quality. Furthermore, additional tailored and rigorous evaluation studies on the mHealth clinical impact and reliability are still required[15,16] . 

In Methods, I would suggest making the first paragraph describing the mobile app clearer. As I continued reading, I understood the app but it is very confusing at the beginning. I would make Fig 2, Fig 1 instead. However, I am most worried about the usability evaluation. Did it happen during the training session, after the simulated emergency consultation? Or some time after? Did the authors measure actual engagement after the training session? I worry that the high scores are influenced by courtesy bias, apart from the selection bias already mentioned by the authors. Besides, the sample size is too small to even speculate about the external validity of the results.

We agree that the mobile app description might appear confusing. We reviewed the drafting and structuring of the section, where we describe the general and functional characteristics of the mobile application. Therefore, we reorganized almost the entirely Mobile application section (Lines 171-227). We included reviewers' recommendations regarding the designation of figures 1 and 2, which we changed (Fig1 instead of Fig 2) in the "Manuscript with track changes." Moreover, the features and functionality of each application module were described separately.

Reviewer's comments regarding the usability evaluation allowed us to include the following text (Lines 237-242) to make more precise the process described in the study procedures section.

"The usability evaluation was carried out in a single session according to the following stages:

1. Delivery of materials (devices, medical records, evaluation forms) and instructions about the application use and evaluation process 

2. Simulated emergency consultation (Interaction with the application) 

3. Individually and confidentially usability evaluation by GPs"

Thank you for pointing out the engagement measure, which is an essential part of the evaluation and was included in the usability evaluation and refers to sections B (user engagement) and F (subjective quality assessment) in the evaluation form (included as Supporting information). 

Although the usability evaluation was performed by each GP anonymously, individually and secretly, we shared the reviewer's concern about the possible introduction of a courtesy bias as a disadvantage of this approach [8]; therefore, we included it as a limitation of our study (Lines 490-493). As a strategy to reduce the risk of courtesy bias, further research could include different assignment strategies for the evaluation, such as the use of digital channels or the use of evaluation tools embedded in the mobile application as the log reporting system.

As a recognized limitation, the sample size and extent of the study should be tackled towards a further stage of clinical impact evaluation. Despite these limitations, study results can be considered as a potential strategy to support GPs in primary healthcare centers, rural areas with a limited internet connection.

In the results:

How was CPG knowledge measure?

In lines 164-165 the authors mention that the majority of the GPs agreed that the app matched the CGP information, but in the previous sentence that almost half actually read them… so that could be also indicating courtesy bias?

The first two questions of the evaluation form qualitatively measured CPG's knowledge (S1 Evaluation form). Nevertheless, through these questions, we gathered a general observation regarding the GP's knowledge level of Ministry of Health’s guidelines for vector-borne disease management. Since our findings revealed a surprising proportion of GPs with a lack of knowledge, we decided to include them as part of the evaluation.

Regarding the GP's perception of the app's content and GCPs agreement, we have clarified in lines 359-361 these results the proportions discriminated by subcategories with the following text:

"The proportion of GPs who reported having more comprehensive knowledge about the guides was 9/20, more than half (5/9) ultimately agreed, and 4/9 partially agreed that the app matched the CPGs."

Based on the evaluation form (S1 Evaluation form) the answer options for the first question included the multiple levels of knowledge about CPGs. The second question's option "Not applicable" was included for those who answered in the first question that they did not know the GCPs, which may have generated confusion among some evaluators. Therefore, we hope that including evaluation form facilitates results interpretation.

In the discussion, the first sentence speaks about the novelty of the study but uses a very restrictive situation so, yes it is novel for that…

We appreciate this comment. We agreed that we present this study under a narrow perspective. However, we wanted to frame the discussion around essential conditions to considerer for the development and evaluation of mobile health interventions by adding the following text in lines 440-446 of the "Manuscript with track changes" file.

"There are relevant conditions to be considered for the mobile health applications development and adoption: the local demographic, geographic, connectivity, culture, and socio-sanitary aspects; and the particular needs of the potential end-users, general public, healthcare workers or surveillance authorities[34]. Among the commercially available mHealth developments, a small proportion combines different functionalities such as vector-borne disease management (evaluation, classification, and treatment) or rapid offline clinical information review."

Also, I do agree that more usability and feasibility studies are needed for mHealth but is this study enough to state this? Is a one-time (not longitudinal) and with a very small sample. This limits the interpretation of the results and I worry about speculations as the one presented in lines 205-207.

We agree that a pilot study approach with the mentioned limitations could also limit the interpretation of results. With the previous text, we intended to narrow the scope of our results and suggested the potential (Lines 416-420) of FeverDx as a support tool susceptible to improvement regarding functionality throught iterative evaluation. We added the following text:

“Hence, the use of FeverDx in regions where arboviruses are endemic or hyperendemic potentially could help GPs and stakeholders in the management of acute febrile syndromes such as arboviruses infections. Although this is a pilot study, our findings are a starting point for the development of new implementation studies that allow the evaluation of application performance in different contexts and scenarios”.

In lines 219, the authors may have omitted other apps. Of the top of my head I can think of Kidenga, for example. What are the other 26 apps relevant to surveillance referring to?

We have added relevant information regarding mobile health developments vector-borne diseases related that we omitted in the revised manuscript. We mentioned the apps related to dengue, chikungunya, and Zika surveillance and some of their features (Lines 432-434). Besides, in lines 403-407, we included commercially available mHealth from the most popular online stores (Google Play Store and Apple Store) to complement the description with the following text (Lines 435-439):

"Among the commercially available mHealth developments, a small proportion combines different functionalities such as vector-borne disease management (evaluation, classification, and treatment) or rapid offline clinical information review." 

Lines 236-237 seem to talk about the general public but this app is not intended for that.

We appreciate this observation, and we agree that this statement might appear out of context. We changed the original statement in lines 469-471, and included the following text: 

"Finally, mobile technology offers unique opportunities to reach and engage healthcare professionals to enhance health literacy and public health authorities for real-time disease surveillance." 

Minor:

Lines 145-147: it is not very clear how the evaluation form was given in this sentence and Table 1 is presented in Results. I would suggest including the evaluation form as supp. material and improve this sentence.

Regarding the use of the evaluation form, in the mentioned lines, we described the components of the evaluation form. As the reviewer suggested, we rewrite the description in lines 333-345 to offer a more accurate description. The evaluation form will be included as supporting information to clarify the description of the evaluation form components. In lines 237-240 (Study procedures section), we described the process in which the evaluation form is given to the evaluators.

Lines 150-151: I assume that the authors referred to median and IQR as is stated in the following paragraph.

In line 347 we referred to median and IQR

Line 183: “as (un)acceptable”?

Lines 391-393 (Manuscript with track changes file) in the results section refers to the evaluation form, B section (user engagement). Question 13 inquiry about the application contents and its suitability for the group of users it is intended. The options for the answer are totally unappropriated, mostly unappropriated, acceptable (selected by one GP), adequately oriented, correctly targeted. This is clarified with the inclusion of the evaluation form.

6. PLOS authors have the option to publish the peer review history of their article (what does this mean?). If published, this will include your full peer review and any attached files

Do you want your identity to be public for this peer review? For information about this choice, including consent withdrawal, please see our Privacy Policy.

Reviewer #1: No

---

## [Decision Letter · Decision Letter 1]

18 Mar 2020

PONE-D-19-31941R1

Acceptability and usability of a mobile application for management and surveillance of vector-borne diseases in Colombia: An implementation study

PLOS ONE

Dear Dr Rodriguez,

Thank you very much for submitting your manuscript "Acceptability and usability of a mobile application for management and surveillance of vector-borne diseases in Colombia: An implementation study" (#PONE-D-19-31941R1) for review by PLOS ONE. As with all papers submitted to the journal, your manuscript was fully evaluated by academic editor (myself) and by independent peer reviewers. The reviewers appreciated the attention to an important health topic, but they raised substantial concerns about the paper that must be addressed before this manuscript can be accurately assessed for meeting the PLOS ONE criteria. Therefore, if you feel these issues can be adequately addressed, we invite you to submit a revised version of the manuscript that addresses the points raised during the review process. We can’t, of course, promise publication at that time.

We would appreciate receiving your revised manuscript by May 02 2020 11:59PM. To enhance the reproducibility of your results, we recommend that if applicable you deposit your laboratory protocols in protocols.io, where a protocol can be assigned its own identifier (DOI) such that it can be cited independently in the future. For instructions see: http://journals.plos.org/plosone/s/submission-guidelines#loc-laboratory-protocols

We look forward to receiving your revised manuscript.

Kind regards,

Abdallah M. Samy, PhD

Academic Editor

PLOS ONE

**Reviewers' comments:**

Reviewer's Responses to Questions

**Comments to the Author**

1. If the authors have adequately addressed your comments raised in a previous round of review and you feel that this manuscript is now acceptable for publication, you may indicate that here to bypass the “Comments to the Author” section, enter your conflict of interest statement in the “Confidential to Editor” section, and submit your "Accept" recommendation.

Reviewer #1: All comments have been addressed

Reviewer #2: (No Response)

2. Is the manuscript technically sound, and do the data support the conclusions?

Reviewer #1: Partly

Reviewer #2: Partly

3. Has the statistical analysis been performed appropriately and rigorously? 

Reviewer #1: Yes

Reviewer #2: N/A

4. Have the authors made all data underlying the findings in their manuscript fully available?

Reviewer #1: Yes

Reviewer #2: No

5. Is the manuscript presented in an intelligible fashion and written in standard English?

Reviewer #1: Yes

Reviewer #2: Yes

6. Review Comments to the Author

Reviewer #1: The authors have responded to my concerns in a satisfactory way and although my concerns about bias and external validity remains, they have been able to make those limitations more explicit. They have greatly improved the manuscript, although minor changes need to be addressed before I can recommend it for publication.

Abstract

1- In the background it says: “A large number of mHealth tools are available; however, very few have been evaluated regarding usability and acceptability.”

This is not appropriate as discussed in the previous review and addressed in the reviewer’s comments. Look at the second paragraph of your introduction and use that argument. Why mhealth can help address the problem.

2- In the results is says “Seventy-five percent of the evaluators reported being aware of the Colombian Ministry of Health’s guidelines.”

This is not in the results though, either include in the results or only indicate that less than half had comprehensive knowledge (see comments on the CPG knowledge section of results).

3- In the conclusion it says “Despite a large number of mHealth tools available, the literature lacks evaluated and evidence-based mobile technology.” This is not completely true as described previously. The new discussion section address this pretty well, so you need to revisit that argument here and match the discussion.

4- Conclusions section is too general, you need to answer why FeverDx could help advance or improve practice based on the results of the pilot study. Also briefly mention the limitations (e.g., although it was a pilot study with a small sample size, FeverDx…)

Introduction

5- Where it says “…time and ease of use, access to content and particularly in a critical step of the reporting process of mandatory notifiable diseases.” Change for “…time and ease of use, and access to content in a critical step of the reporting process of mandatory notifiable diseases.”

6- Where it says “This particular feature, we hypothesize that could potentially improve the currently manual registration and report process, mostly using physical resources (paper forms, physical files)[10].” Which feature are you referring to? Maybe rephrase it as: “We hypothesize that these features (ease of use and rapid access to content) can potentially…”

7- Where it says “There are a recently increasing supply and demand of mobile applications development in the healthcare area”. Change for “Supply and demand of mobile applications development in the healthcare area have been recently increasing.”

8- Where it says “This study evaluates the usability and acceptability of FeverDx…” Briefly mention who valuated it and where. Use the first sentence from Methods.

Methods

9- Where it says “Moreover, an infectious disease specialist reviewed the included recommendations.” Moreover can and should be removed.

10- Where it says “Obtained data from registration, evaluation and reporting is storage” Change storage for stored.

11- The following phrase is confusing: “The evaluation form comprises a 25 item measure that includes four objective quality - information quality, engagement, functionality, aesthetics, and two subjective subscales-quality and impact.”

All items are subjective in the strict sense, based on the S1 File provided, I would recommend that you rephrase it as:

“The evaluation form comprises 25 questions measuring quantity and quality of information (Section A), engagement (Section B), functionality (Section C), aesthetics (Section D), impact of patient management from the practitioner perspective (Section E) and acceptability by health practitioners (Section F) (S1 file).”

12- Where it says “The evaluation test is scored by determining the mean scores of the app quality subscales and the total mean score.” You should rephrase it as: “A global score was obtained for the evaluation form test by determining the median scores of each subsection.”

13- S1 file: I suggest some changes:

Seccion E: the brief description is the same as Seccion D. This section measures impact on patient management from the practitioner perspective.

Seccion F: This section measures acceptability of users. Include this in the description.

Results

14- In the CPG knowledge section, to improve readability, I suggest rephrasing the whole section as:

“The proportion of GPs with comprehensive knowledge of the Colombian’s CPG for diagnosis and management of arboviruses (dengue, chikungunya, and Zika) was less than half of GPs evaluated (9/20) and less than one third of them applied them in their practice (6/20). Of those GPs who reported having comprehensive knowledge of CPG, more than half (5/9) wholly agreed and, 4/9 partially agreed that the application information matched with CPG information.”

15- Table 1: change subjective assessment of quality by acceptability

16- Where it says “Regarding the "interest generated by the application" item, GPs scored a median of 5 (IQR=4-4.5).”, which section is this in the table? Be consistent with names. Also the median doesn’t match the IQR, check that.

17- Where it says “Moreover, all the GPs recommended the app”, change it for “Moreover, all the GPs would recommend the app”

Discussion

18- Where it says “Mobile applications designed for epidemiological surveillance of arboviruses are scope limited”, rephrase it as: “Mobile applications designed for epidemiological surveillance of arboviruses are limited in their scope”

19- Where it says ”On the one hand, a study conducted in 2018”, “On the one hand” is not appropriate. You should remove it.

20- Where it says “...that aimed case reporting and outbreaks..”, change for aimed at case…

21- Where it says “…some limited by country, the language with narrow applicability…” Do you mean that the language limited their applicability because they were mostly in English? Were they designed to be used by English-speaking users or in developed countries? Making that more explicit will help your argument.

22- Where it says “However its implementation in clinical scenarios is scarce”, do you mean they are not designed to be used by GPs in clinical scenarios? Rephrase accordingly.

23- Where it says “A striking find was”, change for “A striking finding was”

24- This phrase “…although the majority of the GPs agreed that the application content was according to management guidelines.” is not necessary here and interrupts the flow. Moreover, using although implies that is in contrast with the previous statement which is not. I would remove this phrase altogether.

Limitations

25- Consider including a statement about their applicability mostly restricted to Colombia, given the country specific CPG and surveillance systems.

Reviewer #2: Even though the article is clear and the mobile app is interesting the findings does not correspond to an orignial article, it is more suitable for a short communication.

Abstract: in methods I would add the grade scale of uMARS, if not results are not understandable without reading the full article.

Introduction: it is not clear which is the potential impact of the using of the app among GP. Line 67, this paragraph should be at the end of the introduction or even in methods.

Methods:

Fig2, when referring to alarm signs that includes all dengue warning signs? if so it should be stated.

Results: CPR knowledge prior of after using the app?

It is not clear if the app was used in clinical context, with how many patients, if it was useful for the final diagnosis.

Discussion: there is no future plans for further research or for the app.

General thoughts: First of all, it could be missleading using an app for fever which do not discriminate malaria or bacterial infections at the begining. An algorithm of fever in the tropics should rule out the most dangerous causes first, and if it is not a malaria zone it should be stated; nevertheless bacterial infections are distributed worlwide.

Although the info about this app should be shared with the community little information could be extracted from this article. I would suggest to compare two GP groups, one using the app and the other not using it and seeing the differences; but the most important thing is that the app is not validated with confirmed cases, apparently the purpose of the app is only to apply the guidelines, if it is so it should be explained. Could be much more interesting if this app can be used by health care workers who not are doctors. Maybe the app should focuse more in epidemiological vigilance.

7. PLOS authors have the option to publish the peer review history of their article (what does this mean?). If published, this will include your full peer review and any attached files.

Reviewer #1: No

Reviewer #2: No

---

## [Author Response · Author response to Decision Letter 1]

28 Apr 2020

Reviewers

PLOS ONE 

April 28th, 2020

Dear reviewers, 

We thank you for a thorough revision and valuable suggestions regarding our manuscript and for the opportunity to revise and resubmit. We are pleased to submit the improved manuscript. On the following pages, we include our response to each recommendation and comments. On behalf of my co-authors, I thank you for your consideration of this resubmission. We appreciate your time and look forward to your response.

Sincerely, 

Sarita Rodríguez R, M.D.

Corresponding author

sarita.rodriguez@fvl.org.co

+573173820407

---

## [Editor Report · Decision Letter 2]

4 May 2020

Acceptability and usability of a mobile application for management and surveillance of vector-borne diseases in Colombia: An implementation study

PONE-D-19-31941R2

Dear Dr. Rodriguez,

We are pleased to inform you that your manuscript, "Acceptability and usability of a mobile application for management and surveillance of vector-borne diseases in Colombia: An implementation study" (PONE-D-19-31941R2), has been judged scientifically suitable for publication and will be formally accepted for publication once it complies with all outstanding technical requirements.

With kind regards,

Abdallah M. Samy, PhD

Academic Editor

PLOS ONE

---

## [Editor Report · Acceptance letter]

13 May 2020

PONE-D-19-31941R2 

Acceptability and usability of a mobile application for management and surveillance of vector-borne diseases in Colombia: An implementation study 

Dear Dr. Rodríguez:

I am pleased to inform you that your manuscript has been deemed suitable for publication in PLOS ONE. Congratulations! Your manuscript is now with our production department. 

With kind regards,

on behalf of

Dr. Abdallah M. Samy 

Academic Editor

PLOS ONE